

# LOGIC: LLM-originated guidance for internal cognitive improvement of small language models in stance detection

Woojin Lee[1],*, Jaewook Lee[1],* and Harksoo Kim[2]

[1] Department of Artificial Intelligence, Konkuk University, Seoul, Republic of South Korea
[2] Department of Computer Science and Engineering, Konkuk University, Seoul, Republic of South Korea
* These authors contributed equally to this work.

## ABSTRACT

Stance detection is a critical task in natural language processing that determines an author's viewpoint toward a specific target, playing a pivotal role in social science research and various applications. Traditional approaches incorporating Wikipedia-sourced data into small language models (SLMs) to compensate for limited target knowledge often suffer from inconsistencies in article quality and length due to the diverse pool of Wikipedia contributors. To address these limitations, we utilize large language models (LLMs) pretrained on expansive datasets to generate accurate and contextually relevant target knowledge. By providing concise, real-world insights tailored to the stance detection task, this approach surpasses the limitations of Wikipedia-based information. Despite their superior reasoning capabilities, LLMs are computationally intensive and challenging to deploy on smaller devices. To mitigate these drawbacks, we introduce a reasoning distillation methodology that transfers the reasoning capabilities of LLMs to more compact SLMs, enhancing their efficiency while maintaining robust performance. Our stance detection model, LOGIC (LLM-Originated Guidance for Internal Cognitive improvement of small language models in stance detection), is built on Bidirectional and Auto-Regressive Transformer (BART) and fine-tuned with auxiliary learning tasks, including reasoning distillation. By incorporating LLM-generated target knowledge into the inference process, LOGIC achieves state-of-the-art performance on the VAried Stance Topics (VAST) dataset, outperforming advanced models like GPT-3.5 Turbo and GPT-4 Turbo in stance detection tasks.

# INTRODUCTION

Stance detection is a critical natural language processing task that discerns whether a text author expresses a positive, negative, or neutral stance toward a specific target (*Somasundaran & Wiebe, 2010*; *Augenstein et al., 2016*; *Mohammad et al., 2016*; *Sobhani, Inkpen & Zhu, 2017*; *Wen & Hauptmann, 2023*). This task is crucial for understanding different perspectives in various forms of media, such as social media posts and news articles (*Umer et al., 2020*; *Conforti et al., 2020*; *Glandt et al., 2021*; *Li et al., 2021*). The target in the text can be explicitly mentioned or indirectly implied, making it a pivotal

Corresponding author
Harksoo Kim,
nlpdrkim@konkuk.ac.kr

**Table 1 A stance detection dataset example from VAST.**

| | |
|---|---|
| Target | Electric car |
| Text | Well beyond doubt, the Electric Car technology is an environment friendly technology and a welcome one. But when it comes to business, it is not only the future savings, but also the affordability factor that needs to be kept in mind. As usual, only a select few will go for it initially. Later the masses will follow the suit, as and when the products get more and more affordable. |
| Stance | Positive |

element. Understanding the relationship between the target and the text is essential to accurately identify the stance. Table 1 shows an example from the VAried Stance Topics (VAST) dataset (*Allaway & McKeown, 2020*), illustrating how the target, text, and stance are connected, which are key elements in stance detection.

Early research in stance detection primarily focused on target-specific stance detection, developing models for specific targets but with limitations in predicting the stance of unseen targets (*Hasan & Ng, 2014*; *Zhou, Cristea & Shi, 2017*; *Mohtarami et al., 2018*). To address these challenges, subsequent studies investigated cross-target stance detection (*Zhang et al., 2020*; *Liang et al., 2021*), aiming to extend the model's applicability by predicting the stances of related targets based on knowledge of specific targets. However, cross-target stance detection is fundamentally limited by the need for extensive datasets encompassing various targets for effective generalization. Recent research trends have shifted toward zero-shot and few-shot stance detection (*Allaway, Srikanth & McKeown, 2021*; *Jiang et al., 2022*). These approaches aim to broaden model applicability by enabling accurate stance prediction for targets not seen during training. Until recently, these approaches were primarily based on classification methods. In recent work, *Wen & Hauptmann (2023)* redefined stance detection as a conditional generation task, eliminating the need for an additional classification layer. Their model integrates Wikipedia knowledge directly into the inputs of small language models (SLMs) like Bidirectional and Auto-Regressive Transformer (BART) (*Lewis et al., 2019*) to predict stance labels directly. However, this method heavily relies on the quality and relevance of Wikipedia knowledge and the reasoning ability of SLMs.

The limitations of Wikipedia-based knowledge include variability in article length, which often exceeds the maximum input token length that SLMs can process. Additionally, some targets lack a corresponding Wikipedia article or have insufficient information. Traditional SLMs also have relatively weaker reasoning power compared to current large language models (LLMs). Although directly employing LLMs for inference seems viable due to their strong reasoning capabilities, their vast parameter size leads to significant computational costs and resource consumption, limiting scalability on smaller devices and posing deployment challenges. In this article, we address these challenges by developing a strategy that leverages LLMs to enhance stance detection through two main components:

**Target knowledge extraction.** We use LLMs to generate concise and contextually relevant target knowledge that reflects real-world contexts, surpassing the limitations of

Wikipedia-sourced data. This knowledge is then integrated into our model inputs to improve stance detection accuracy.

**Reasoning distillation.** Instead of directly using resource-intensive LLMs for inference, we employ reasoning distillation to transfer LLM-level reasoning capabilities to SLMs. We generate reasoning texts using LLMs for training dataset samples with known answers, and these texts serve as auxiliary learning guidance to effectively distill reasoning power. This methodology allows SLMs to maintain computational efficiency while benefiting from deeper insights.

Our stance detection model, LOGIC (LLM-Originated Guidance for Internal Cognitive improvement of small language models in stance detection), built on BART and fine-tuned with auxiliary learning tasks, achieves state-of-the-art performance on the VAST dataset, outperforming advanced models like GPT-3.5 Turbo and GPT-4 Turbo in stance detection task. our main contributions are summarized as follows:

- We propose a novel approach that leverages LLMs to generate accurate and contextually relevant target knowledge, surpassing the limitations of Wikipedia-sourced data and providing concise, real-world insights tailored to stance detection tasks.
- We introduce a reasoning distillation methodology that transfers the superior reasoning power of LLMs to SLMs, maintaining computational efficiency while benefiting from deeper insights.
- Our proposed model, LOGIC, achieves new state-of-the-art results on the VAST dataset and outperforms GPT-3.5 Turbo and GPT-4 Turbo in zero-shot and few-shot stance detection tasks.

## RELATED WORK

### Zero-shot and few-shot stance detection

Recent advancements in stance detection have focused on zero-shot and few-shot stance detection, where the goal is to accurately determine stances towards previously unseen topics with minimal training data. A significant contribution in this area is the VAST dataset built by *Allaway & McKeown (2020)*, which offers diverse topics and introduces a model that captures implicit target relationships. Efforts to enhance stance detection have also involved integrating common-sense knowledge graphs (*Liu et al., 2021*) and enriching models with Wikipedia content for better target comprehension (*He, Mokhberian & Lerman, 2022*). In parallel, contrastive learning methodologies have shown promise in zero-shot and few-shot stance detection. *Liang et al. (2022a)* applied hierarchical contrastive learning methods, and *Liang et al. (2022b)* introduced joint contrastive learning approaches. Additionally, *Hanley & Durumeric (2023)* developed topic-agnostic and topic-aware embeddings using contrastive learning with large-scale news datasets.

A notable advancement was made by *Wen & Hauptmann (2023)*, who redefined stance detection as a conditional generation task, leveraging target knowledge from Wikipedia to improve label representation. This framework eliminates the need for an additional

classification layer. Our approach draws inspiration from *Wen & Hauptmann (2023)*, specifically from their use of auxiliary learning tasks (*e.g.*, target prediction, unlikelihood training), but distinguishes itself by integrating target knowledge extracted from LLMs directly into the model's inputs and utilizing LLM-generated reasoning data for reasoning distillation as an auxiliary training task.

## Reasoning distillation from LLMs

LLMs have demonstrated impressive capabilities across various tasks due to their extensive knowledge base and reasoning power. For instance, LLMs like GPT-3.5 (*Brown et al., 2020*), Med-PaLM (*Singhal et al., 2023*), and GPT-4 (*OpenAI, 2023*) have excelled in medical question-answering tasks, such as the United States Medical Licensing Examination (USMLE), significantly surpassing the passing score (*Kung et al., 2023*; *Nori et al., 2023*). However, deploying LLMs in offline and privacy-sensitive environments remains challenging due to their black-box nature and high computational costs. Thus, alternative solutions are needed to leverage the capabilities of LLMs while maintaining efficiency and privacy in such settings.

Recent works have attempted to distill the reasoning ability of LLMs into SLMs (*Magister et al., 2023*; *Yang et al., 2023*; *Shen et al., 2023*). Reasoning ability, as an emergent property of intelligence, encompasses the capacity to interpret information, identify patterns, draw logical conclusions, and make decisions (*Johnson-Laird, 1983*). This ability enables LLMs to excel across a range of reasoning tasks, including numerical reasoning, symbolic reasoning, and understanding implicit relationships within text. A technique for using this reasoning ability to solve inference problems is chain-of-thought (CoT) prompting, which improves reasoning performance by guiding the model to solve reasoning problems step by step (*Wei et al., 2022*; *Kojima et al., 2022*). *Shen et al. (2023)* used CoT prompts to generate rationales that guide student models in solving complex reasoning tasks. *Magister et al. (2023)* demonstrated that specialized prompts could effectively transfer a larger teacher model's reasoning capabilities to a smaller student model, enhancing its performance in various reasoning tasks.

However, this approach faces limitations as generating reasoning data for distillation using models at the scale of hundreds of billions of parameters, such as PaLM-540B (*Chowdhery et al., 2022*) and GPT-3 (*Brown et al., 2020*), requires substantial computational resources. CoT-based methods, while effective, demand significant resources due to the complexity of generating multi-step reasoning sequences, making data generation costly, particularly in large-scale models.

In response, recent studies have sought methods to reduce resource consumption while maintaining reasoning quality. For example, *Kang et al. (2024)* proposed a method to distill the reasoning capabilities of LLMs into smaller models like 250M T5 (*Raffel et al., 2020*), while also injecting domain-specific knowledge. Their framework incorporates a retriever to obtain relevant passages from external knowledge bases (*e.g.*, Wikipedia) to generate rationales. The small LM is then fine-tuned to generate reasoning based on both the

question and the retrieved documents, and predict the answer. However, Knowledge-Augmented Reasoning Distillation (KARD) still relies on external knowledge sources, which poses a limitation. Stance detection, being closer to logical judgment than knowledge-intensive tasks, may not be directly suitable for applying such methodologies.

To address these limitations, our approach aims to reduce the cost of reasoning generation by employing optimized single prompts without CoT, while also demonstrating that effective reasoning for stance detection can be generated using only the internal knowledge of LLMs, without relying on external knowledge bases. By distilling this reasoning into smaller models, we aim to significantly enhance their reasoning capabilities, achieving substantial performance improvements in stance detection.

Our work builds upon these findings by employing specialized prompts that better capture the author's intent and by categorizing the reasoning process into 'Short' and 'Long' categories. 'Short' reasoning provides concise logical explanations within three lines for quick and intuitive decisions, while 'Long' reasoning includes a wealth of detailed knowledge and explanations for in-depth analysis. This comprehensive strategy enables our model to effectively transfer LLM-level reasoning capabilities to smaller language models while maintaining computational efficiency.

## METHODOLOGY

In this study, we investigated a template-based generation approach for stance detection and explored various auxiliary learning tasks to enhance the model's understanding of deeper semantic relationships. We also discuss how the LLM can be used to generate additional data and integrate this information as auxiliary datasets that expand the original dataset. By providing LLM-generated target knowledge to enhance the model's understanding of the target during inference, and serving LLM-generated reasoning as auxiliary training data for the auxiliary task of reasoning generation, we aim to improve the model's reasoning capabilities specifically for stance detection. The complete framework of this study is shown in Fig. 1.

### Template-based generation approach

Taking inspiration from *Wen & Hauptmann (2023)*, we adopted a template-based generation approach for stance detection. Our LOGIC model takes a partially filled template as input, comprising two sentences: "Topic is *<topic>*. Stance is *<stance>*." In this template, placeholders are pre-filled with the correct values, except for the portion targeted for prediction.

In this study, we employed BART (*Lewis et al., 2019*) as a template-based generation approach. BART is a pretrained language model with an encoder-decoder architecture that is pretrained with denoising objectives. BART text generation is based on predicting the next token to be generated based on the tokens generated thus far:
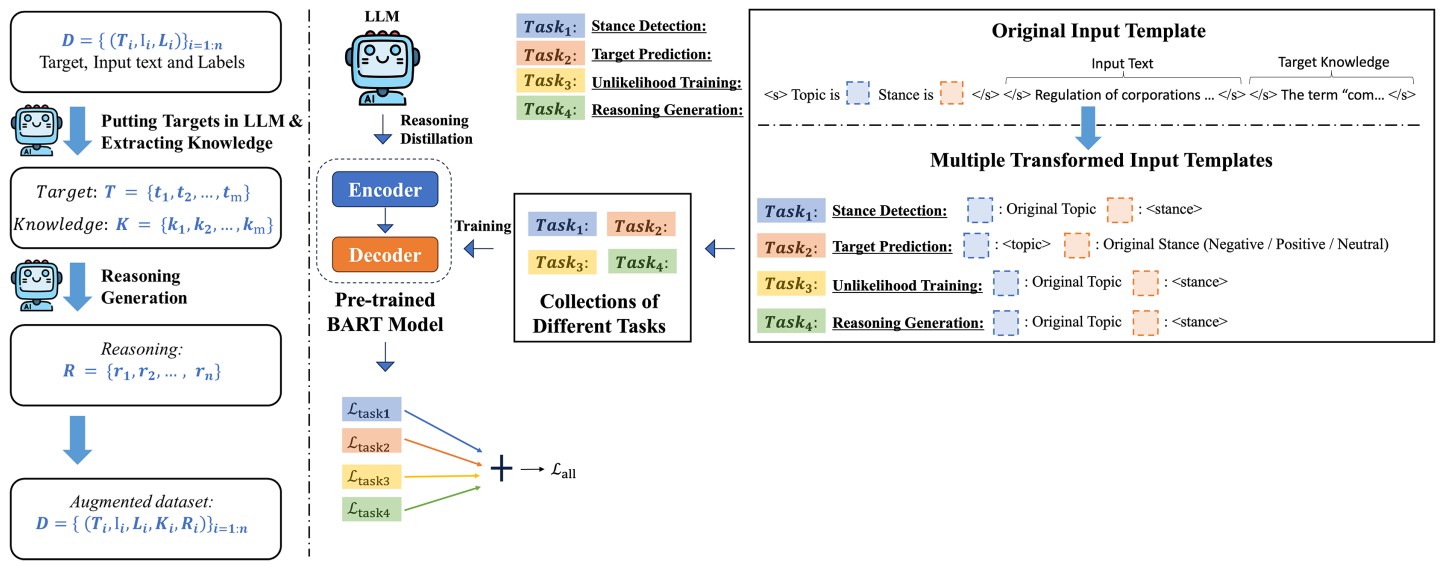

**Figure 1** Our general pipeline for target knowledge extraction and reasoning generation using LLM and our LOGIC model training framework overview.

$$p(o \mid g(x, t); \theta) = \prod_{i=1}^{|o|} p(o_i | o_{<i}, g(x, t); \theta), \tag{1}$$

$o$ represents the final output, symbolizing a fully-filled template. The function $g(x, t)$ generates a template used as input, structured as "$<s>$ template $</s> </s> x </s>$", where $x$ denotes the input text following the partially filled template mentioned earlier. $t$ represents the target and substitutes the $<topic>$ placeholder in the template. The $<stance>$ placeholder is used unchanged, and the model predicts the polarity word that fits in the $<stance>$ placeholder. In this process, the model is trained to minimize the log-likelihood. The loss function for stance detection is as follows:

$$\mathcal{L}_s = -\sum_{i=1}^{|O|} log\ p(o_i | o_{<i}, g(x, t); \theta). \tag{2}$$

## Auxiliary learning tasks

**Target prediction.** A profound comprehension of the relationship between the targets and text is crucial for stance detection performance. To enhance our model's understanding, we converted the stance detection task into a target prediction task and adopted it as an auxiliary learning task. To do this, we insert the actual polarity word into the $<stance>$ position of the existing template and leave the $<topic>$ position unchanged. The model is trained in the similar way as for stance detection. The model underwent training similar to that used for stance detection. The model first learns by taking partially filled inputs and filling in the $<topic>$ position. The loss function used for the target prediction is as follows:

$$\mathcal{L}_t = -\sum_{i=1}^{|O|} log \ p(o_i | o_{<i}, g(x, s); \theta). \tag{3}$$

**Unlikelihood training.** In our stance detection model, we employ a log-likelihood objective to enhance the probability of generating a suitable polarity word for the *<stance>* position. However, considering the existence of multiple potential candidates, it is essential to diminish the probability of generating an unsuitable option. To tackle this challenge, we introduced unlikelihood training (*Welleck et al., 2019*) as an auxiliary learning task.

This task utilizes a stance detection template as input but alters the polarity label of the output value *o* by substituting it with an unlikelihood token. The modified output is represented as $o'$, where $o_k$ and $o'_k$ represent the polarity labels. The loss function used in the unlikelihood training is as follows:

$$\mathcal{L}_u = -\underbrace{\sum_{i \neq k} log \ p(o'_i | o'_{<i}, g(x, t); \theta)}_{likelihood}$$
$$-\alpha \cdot \underbrace{log(1 - p(o'_k | o'_{<k}, g(x, t); \theta))}_{unlikelihood}, \tag{4}$$

$\alpha$ is an adjustable hyperparameter that is set to 1. All other hyperparameters mentioned in the following sections were similarly optimized through empirical experimentation in this study.

**Reasoning generation.** Unlike previous works, our LOGIC model not only predicts stances but also generates logical reasoning that elucidates the semantic relationships between the target, input text, and stance. As the original dataset did not have reasoning data, we utilized GPT-3.5 Turbo to generate this additional data. Our goal was to distill the comprehensive reasoning capabilities of GPT-3.5 Turbo into our smaller model. Detailed information on the generation of reasoning data is provided in the following subsection.

The input for reasoning generation mirrors that of stance detection. Initially, the model predicts the polarity words to fit the *<stance>* placeholder in the input template. Following this, it generates reasoning explanations. Short reasoning provides a concise and direct explanation, whereas long reasoning offers a detailed and comprehensive analysis. The final output of the model is: "*<s>*Topic is *<topic>*. Stance is *<stance>*. (*<reasoning_short>* or *<reasoning_long>*) *<reason>*". Here, the *<topic>*, *<stance>*, and *<reason>* placeholders are populated with the corresponding answer values and LLM-generated reasoning. The special tokens *<reasoning_short>* and *<reasoning_long>* distinguish between short and long reasoning types. For additional template details, refer to Fig. 2. The loss function for reasoning generation is expressed as follows:

$$\mathcal{L}_r = -\sum_{i=1}^{|O|} log \ p(o_i | o_{<i}, g(x, t); \theta), \tag{5}$$

|  | Stance Detection | Target Prediction |
|---|---|---|
| Input | Topic is electric car. Stance is *<stance>*. | Topic is *<topic>*. Stance is positive. |
| Output | Topic is electric car. Stance is positive. | Topic is electric car. Stance is positive. |
|  | **Unlikelihood Training** | **Reasoning Generation** |
| Input | Topic is electric car. Stance is *<stance>*. | Topic is electric car. Stance is *<stance>*. |
| Output | Topic is electric car. Stance is negative. | Topic is electric car. Stance is positive. (*<reasoning_short>* or *<reasoning_long>*) The given post expresses a positive attitude towards electric cars. The author acknowledges the environmental benefits of electric car ⋯ |

**Figure 2 Sample input and output templates for stance detection, target prediction, unlikelihood training, and reasoning generation.** The example in this figure corresponds to the sample in Table 1, allowing for a more concrete understanding of how the templates work with the data sample.

**Training object.** The final loss function is the weighted sum of the respective loss functions for stance detection, target prediction, unlikelihood training, and reasoning generation:

$$\mathcal{L} = \mathcal{L}_s + \alpha_t \mathcal{L}_t + \alpha_u \mathcal{L}_u + \alpha_r \mathcal{L}_r \tag{6}$$

We set $\alpha_t$ to 1, $\alpha_u$ to 0.5, and $\alpha_r$ to 3.

## Target knowledge extraction and reasoning generation using LLM

Recent research in stance detection has predominantly relied on Wikipedia articles as an external knowledge source for targets (*He, Mokhberian & Lerman, 2022*; *Wen & Hauptmann, 2023*). However, the quality and length of information in these articles are contingent on the editors' expertise. Additionally, when a Wikipedia article exceeds the maximum input token length for SLMs, its full content cannot be effectively utilized. To overcome these limitations, we propose a methodology for extracting target knowledge from LLMs. We input custom-designed prompts into the LLM to analyze the target, its real-world significance, and its contextual usage. By designing prompts to selectively extract key information crucial for logical reasoning, this methodology yields concise and high-quality target knowledge. Ultimately, it enhances the accuracy and robustness of stance detection by providing a relatively consistent level of quality and length in target knowledge.

While attempts to address the knowledge deficiency in SLMs through external knowledge can enhance performance, these efforts do not provide a fundamental solution to the intrinsic knowledge limitations of SLMs. As an alternative, using LLMs equipped with rich internal knowledge directly addresses this issue. However, this approach poses challenges in terms of computing cost and processing speed, especially on small devices. To resolve this, we employ reasoning distillation to transfer the reasoning power in LLMs to SLMs. This strategy alleviates the knowledge deficiency in SLMs and substantially improves portability to small devices.

| Knowledge Extraction & Reasoning Generation | |
|---|---|
| **Prompt Type 1 : Target Knowledge Extraction**<br>Explain to me the definition of '*{topic}*', what the term means in the real world, and in what context it is used. Please summarize and explain only the most important points.<br>**Prompt Type 2 : Short Reasoning Generation**<br>In the context of the following post and related topic, Explain in 3 sentences why the position '*{stance}*' best represents the attitude expressed in the post toward that topic. Your explanation must be logical and clear in 3 sentences. You must choose '*{stance}*' stance. Your subjective judgment is not permitted. | **Prompt Type 3 : Long Reasoning Generation**<br>In the context of the following post and related topics, explain why the '*{stance}*' stance best represents the attitude expressed in the post about that topic. You must select the given stance '*{stance}*'. Your subjective judgment is not allowed. When generating your response, please start by choosing the exact stance '*{stance}*'. Also, you must explain why it is not '*{false1}*', '*{false2}*' in details. Provide a comprehensive analysis based on the mentioned factors and elucidate why the given stance is suitable for a post related to the topic. |

**Figure 3 Prompt types for target knowledge extraction and short & long reasoning generation.**

The LLM used for knowledge generation is GPT-3.5 Turbo, while BART (140M parameter size) is selected as the SLM due to its portability. Specifically, we generate reasoning data through GPT-3.5 Turbo by analyzing the relationships between the target, the input text, and their interactions represented by labels. This generated reasoning data is utilized to train SLM to predict stances and generate rationale for those predictions. This method transfers the reasoning abilities of LLMs to SLMs, enabling them to not only predict stance but also generate clear rationales for their predictions. The prompts used for data generation are displayed in Fig. 3.

**Target knowledge extraction prompting.** In this study, we devised a method for extracting target knowledge using LLMs. Utilizing a prompt-based approach with the GPT-3.5 Turbo model, we aimed to identify and summarize the definition of the target text, its real-world significance, and its prevalent contextual usage. Specifically, as shown in Fig. 3, the token *<topic>* is used in the target knowledge extraction prompt to represent the target of the stance detection task, guiding the LLM to generate relevant contextual information. This approach guided the model to focus on essential information, thereby improving the overall quality of information by excluding unnecessary details for logical reasoning.

**Reasoning generation prompting.** We introduce a methodology for reasoning generation that aims to distill the LLM's capacity to analyze and interpret the relationship between targets and text into a smaller model. Short reasoning prompts involve analyzing the text's stance within a limited number of sentences, generating concise and logical explanations. In contrast, long reasoning prompts do not limit the length of reasoning; they generate extensive reasoning data that details the rationale for stance detection and analyze why other stance candidates are unsuitable. Specifically, in the context of long reasoning generation, we use tokens like *<stance>*, *<false1>*, and *<false2>* to structure the explanation. Here, *<stance>* represents the correct stance label (*e.g.*, Positive), while *<false1>* and *<false2>* refer to incorrect alternatives (*e.g.*, Negative or Neutral). During this process, the LLM is tasked with not only explaining why the *<stance>* is correct, but also

elaborating on why the alternative labels *<false*1> and *<false*2> do not apply. This allows the model to generate a more comprehensive and nuanced rationale for the stance prediction, strengthening its interpretability and robustness. These prompts were crafted to guide the model in aligning with human-annotated correct answer labels and generating corresponding rationales.

# EXPERIMENTS

## Dataset

The VAST dataset (*Allaway & McKeown, 2020*) is designed explicitly for zero-shot and few-shot stance detection, offering a broad spectrum of topics across fields such as politics, education, and public health. The dataset comprises 18,548 examples and 5,630 targets for stance detection, ensuring linguistic diversity in the wording of targets. The original examples were collected by *Habernal et al. (2018)* under the Apache-2.0 license. The dataset primarily consists of comments gathered from the New York Times' "Room for Debate" section, which are representative of various topics. Each comment typically consists of several sentences, with the length of texts varying depending on the complexity of the topic. Stance labels are crowd-sourced using Amazon Mechanical Turk, where three workers annotate each example. The final label is determined through majority voting, and annotations are validated for quality through manual inspection and MACE (*Hovy et al., 2013*) analysis. To ensure robustness in evaluation, the dataset is divided into training, validation, and test sets, following a standard 70-15-15 split.

For the purpose of evaluation, two types of tasks are considered: zero-shot and few-shot. In contrast to the common definitions of zero-shot and few-shot used in general machine learning inference, the VAST dataset distinguishes these terms specifically for stance detection, as outlined in the introduction. In zero-shot cases, the model is tested on target topics that it has never encountered during training, meaning these topics are not included in the train set. This requires the model to generalize its learning to entirely new subjects. In contrast, in few-shot cases, the model is tested on target topics that it has seen during training, meaning they are included in the train set but with only a limited number of examples. This distinction ensures that the model is challenged to perform well not only on familiar topics with sparse data but also on completely unseen topics.

Following the methodology outlined by *He, Mokhberian & Lerman (2022)*, Wikipedia data was employed as additional input for target knowledge in our comparative experiments, serving a similar function to LLM-generated target knowledge. This data was utilized as a baseline to validate the effectiveness of our proposed approach. By adopting the same approach as *Wen & Hauptmann (2023)* for comparison, we ensure that our LLM-based method, which does not rely on Wikipedia, can be directly and fairly evaluated against the current state-of-the-art. The original Wikipedia dataset was constructed in compliance with the CC-BY-SA license, as specified by *He, Mokhberian & Lerman (2022)*.

## Experimental settings

In this study, we demonstrate the superiority of our LOGIC model for stance detection using the VAST dataset. Our primary focus was on evaluating how well our trained

**Table 2 Performance metrics for the ablation study on the development set of the VAST dataset.** All metrics are shown as the average of the five seed results. SRG and LRG are tasks that train the model on short reasoning generation and long reasoning generation, respectively. ARG is the task that trains both SRG and LRG. The model highlighted in gray in the last row of the table corresponds to our LOGIC model. Bolded values indicate the highest performance for each metric, and this convention is applied to all subsequent tables.

| Models | Zero-shot | | | Few-shot | | | Overall | | |
|---|---|---|---|---|---|---|---|---|---|
| | Precision | Recall | F1 | Precision | Recall | F1 | Precision | Recall | F1 |
| BART$_{base}$ | 76.6 | 76.0 | 76.2 | 77.3 | 76.3 | 76.5 | 76.9 | 76.3 | 76.5 |
| + Target prediction | 78.0 | 77.4 | 77.7 | 77.3 | 76.7 | 76.8 | 77.7 | 77.2 | 77.3 |
| + Unlikelihood training | 78.1 | 77.6 | 77.8 | 77.1 | 76.3 | 76.4 | 77.6 | 77.0 | 77.2 |
| *Reasoning generation (with all existing auxiliary learning tasks)* | | | | | | | | | |
| + SRG | 77.3 | 76.4 | 76.7 | 77.8 | 77.0 | 77.2 | 77.5 | 76.8 | 77.0 |
| - SRG + LRG | 77.3 | 76.1 | 76.4 | 77.7 | 77.0 | 77.2 | 77.5 | 76.6 | 76.9 |
| + ARG | 77.4 | 76.4 | 76.7 | 77.9 | 77.0 | 77.3 | 77.6 | 76.8 | 77.1 |
| *Wikipedia knowledge integration (with all existing auxiliary learning tasks)* | | | | | | | | | |
| + Wikipedia knowledge | 78.7 | 78.1 | 78.3 | 78.5 | 77.6 | 77.7 | 78.6 | 78.0 | 78.2 |
| + Wikipedia knowledge + SRG | 79.3 | 78.4 | 78.7 | 78.9 | 78.5 | 78.7 | 79.1 | 78.6 | 78.8 |
| + Wikipedia knowledge + LRG | 79.5 | 78.8 | 79.0 | 78.3 | 77.6 | 77.8 | 78.9 | 78.4 | 78.6 |
| + Wikipedia knowledge + ARG | 80.3 | 79.5 | 79.7 | 79.4 | 78.3 | 78.5 | 79.8 | 79.0 | 79.2 |
| *LLM target knowledge extraction (with all existing auxiliary learning tasks)* | | | | | | | | | |
| + LLM target knowledge | 80.2 | 79.5 | 79.7 | 79.2 | 78.8 | 78.9 | 79.7 | 79.2 | 79.4 |
| + LLM target knowledge + SRG | 81.2 | 80.4 | 80.6 | 80.4 | 80.0 | 80.1 | 80.8 | 80.3 | 80.4 |
| + LLM target knowledge + LRG | 80.9 | 80.3 | 80.5 | 80.7 | 80.2 | 80.4 | 80.8 | 80.3 | 80.5 |
| + LLM target knowledge + ARG | **81.7** | **81.2** | **81.4** | **81.1** | **80.6** | **80.8** | **81.4** | **81.0** | **81.1** |

model's predicted stance labels matched the actual stance labels in the VAST dataset. To assess this, we employed precision, recall, and F1 score as our evaluation metrics, consistent with previous studies.

We trained the model for 30 epochs on the VAST dataset using an Nvidia RTX 4090 GPU. Due to hardware limitations, we opted for a batch size of eight with four accumulation steps, instead of the desired batch size of 32. To ensure stability and convergence during training, we utilized a linear scheduler with a warm-up ratio set to 0.1, gradually increasing the learning rate. We used a range of hyperparameters, such as a learning rate of 5e−6 and a maximum decoding length of 1,024 tokens. All hyperparameters were empirically determined through multiple trials to achieve optimal performance.

Furthermore, we distinguish between zero-shot and few-shot evaluations. In zero-shot settings, the model is tested on topics it has never seen during training (*i.e.*, topics not included in the train set), which challenges its generalization ability. In contrast, few-shot settings test the model on topics it has seen during training, albeit with limited data. The "Overall Setting" refers to the aggregated results of both zero-shot and few-shot performances, offering a comprehensive measure of the model's ability to handle both familiar and unfamiliar topics.

**Table 3 F1 score (%) illustrating stance detection performance across the VAST test set.**

| Models | VAST | | |
|---|---|---|---|
| | **Zero-shot** | **Few-shot** | **Overall** |
| TGA-Net | 66.6 | 66.3 | 66.5 |
| BERT-GCN | 68.6 | 69.7 | 69.2 |
| CKE-Net | 70.2 | 70.1 | 70.1 |
| JointCL | 72.3 | 71.5 | 71.9 |
| WS-BERT | 75.3 | 73.6 | 74.5 |
| TATA | 77.1 | 74.1 | 75.6 |
| BART$_{generation}$ | 76.4 | **78.8** | 77.3 |
| **LOGIC** | **80.2** | **78.8** | **79.5** |

Note:
Bolded values indicate the highest performance for each metric.

**Table 4 Performance comparison of our LOGIC model with GPT-3.5 Turbo and GPT-4 Turbo on the VAST test set.**

| Models | VAST | | |
|---|---|---|---|
| | **Precision** | **Recall** | **F1** |
| GPT-3.5 Turbo 0-shot | 49.0 | 48.1 | 47.5 |
| GPT-3.5 Turbo 12-shot | 49.0 | 48.8 | 48.6 |
| GPT-3.5 Turbo 24-shot | 53.6 | 53.5 | 53.5 |
| GPT-4 Turbo 0-shot | 69.1 | 65.1 | 65.2 |
| GPT-4 Turbo 12-shot | 66.6 | 62.0 | 61.8 |
| GPT-4 Turbo 24-shot | 67.2 | 61.2 | 61.0 |
| **LOGIC** | **79.7** | **79.4** | **79.5** |

Note:
Bolded values indicate the highest performance for each metric.

Several ablation studies were conducted to assess the impact of the auxiliary learning tasks. When trained on the full dataset with all auxiliary tasks included, the training process took approximately 5 h to complete 30 epochs on the VAST dataset. For random variability and robustness, all experiments were conducted with multiple random seeds, and results are averaged across these runs to account for fluctuations in performance due to initialization.

Additionally, while Table 2 presents the development set performance, Tables 3 and 4 report the test set performance for a more rigorous comparison. The reasoning behind this distinction is rooted in the purpose of the ablation studies and model evaluation. In ablation studies, the focus is typically on fine-tuning and evaluating how different components of the model contribute to performance. For this purpose, the development set is used to iteratively adjust the model, as it allows us to optimize hyperparameters and refine model configurations without risking overfitting on the test set. The results on the development set help determine the most effective combination of methods before final evaluation. On the other hand, the test set is reserved for final comparisons against other

models. Using the test set ensures that our model's performance is rigorously evaluated in a more generalized and unbiased manner, as the test set is completely unseen during the training and tuning process. This separation ensures that the results reported in the final comparisons reflect the model's true ability to generalize to new data, providing a fair basis for benchmarking against other models.

In terms of raw data, we avoided excessive preprocessing to retain the integrity of the original data. Our code directly accessed the raw VAST dataset in the original CSV format provided by *Allaway & McKeown (2020)*, and the Wikipedia dataset from *He, Mokhberian & Lerman (2022)* was used in its original Pickle (PKL) format. For LLM-generated target knowledge and reasoning, we treated these as auxiliary datasets. Specifically, the LLM-generated reasoning data was appended as additional columns in the VAST CSV file, while the LLM target knowledge followed the same PKL format as the Wikipedia dataset. This approach ensured a seamless integration of both original and auxiliary datasets into our model pipeline.

## Impact of existing auxiliary learning tasks and reasoning generation task

In this section, we assess the influence of current auxiliary learning and reasoning generation tasks on stance detection performance. The first row block in Table 2 illustrates the performance change when the model undergoes training with each auxiliary learning task. All performance metrics are averaged across five different seeds in the development set. $BART_{base}$ is trained only on the stance detection task. We demonstrate that training the model with existing auxiliary tasks, such as target prediction and unlikelihood training, enhances the stance detection performance. This improvement was observed compared with $BART_{base}$. Specifically, target prediction proves advantageous in few-shot scenarios, while unlikelihood training excels in zero-shot scenarios.

The second row block shows the performance when the model is trained with our proposed reasoning generation task in addition to all the auxiliary learning tasks in the first row block. Training the model with reasoning data generated by the LLM improved the overall performance compared with $BART_{base}$. In zero-shot and few-shot scenarios, short reasoning generation (SRG) improves F1 scores by 0.5 and 0.7 points, respectively. Likewise, long reasoning generation (LRG) improves F1 scores by 0.2 points in zero-shot and 0.7 points in few-shot scenarios. All reasoning generation (ARG), which incorporates both short and long reasoning, leads to the highest overall improvement, by 0.5 points in the zero-shot scenario and 0.8 points in the few-shot scenario.

However, the model additionally trained on ARG exhibited a marginal decrease in the overall F1 score compared to the model trained solely on all existing auxiliary learning tasks. This slight dip is attributed to the model additionally trained with ARG performing 1.1 points less effectively in the zero-shot scenario compared to the model trained solely on all existing auxiliary learning tasks. However, in the few-shot scenario, the model additionally trained with ARG outperformed the model relying solely on all existing auxiliary learning tasks, demonstrating an improvement of 0.9 points. This suggests that

reasoning generation helps the model understand the deeper relationships between the target, input text, and labels, and provides a performance advantage in few-shot scenarios.

## Analyzing the impact of incorporating external knowledge

**Synergy of Wikipedia target knowledge and reasoning generation.** The experimental results presented in the third block of Table 2 highlight the synergy achieved by combining Wikipedia's target knowledge with reasoning generation tasks. Incorporating Wikipedia target knowledge into the model's input alone enhances performance, but even greater gains are observed when the model is additionally trained in the reasoning generation tasks. Specifically, the combination of Wikipedia knowledge with SRG or LRG improves the F1 scores in both zero-shot and few-shot scenarios.

The combination of Wikipedia knowledge and ARG, which incorporates both SRG and LRG, yields the highest performance improvement. This combination results in a 1.4 points increase in F1 score in the zero-shot scenario and a 0.8 points increase in the few-shot scenario, leading to an overall performance improvement of 1.0 points. SRG emphasizes concise and direct analysis, while LRG focuses on more complex and detailed analyses. Consequently, ARG, which combines both SRG and LRG, trains the model with reasoning of varying lengths and complexities, enabling it to operate most effectively.

The results of combining external knowledge with reasoning generation tasks contrast with the findings in last subsection. Training with reasoning generation tasks alone was beneficial primarily in the few-shot scenario. However, when external knowledge is incorporated into the model's input and the model is trained with reasoning generation tasks, significant performance improvements are observed across all scenarios. This is because the external knowledge, integrated into the encoder along with the model's input text, provides the necessary context for reasoning generation. This integration allows the model to deepen its understanding of the input text and make more precise inferences about complex situations, leading to an overall increase in performance.

**Synergy of LLM target knowledge and reasoning generation.** This study conducted an in-depth analysis of the efficacy of integrating LLM target knowledge with reasoning generation to train the model. As shown in the last block of Table 2, models leveraging LLM target knowledge exhibit significantly higher performance than those using other approaches. This enhancement is consistent across both zero-shot and few-shot scenarios, demonstrating the substantial impact of LLM target knowledge on stance detection.

The reason for the superior performance of LLM target knowledge lies in its ability to provide comprehensive, contextually rich information that SLMs often lack. By integrating this knowledge, the model gains a deeper understanding of the target, which is crucial for accurate stance detection. This is particularly beneficial in scenarios where the target is not explicitly mentioned or is indirectly implied.

Additionally, our experiments confirmed that incorporating reasoning data further enhances the performance of models using LLM target knowledge. Reasoning generation tasks, such as SRG and LRG, add another layer of depth by training the model to generate coherent and contextually appropriate reasoning based on the input. Specifically, both

SRG and LRG showed notable improvements in the overall F1 score, by 1.0 and 1.1 points respectively. These improvements were evident in both zero-shot and few-shot scenarios.

The most significant performance boost was achieved with ARG, which combines SRG and LRG. This approach led to an overall score increase of 1.7 points, with enhancements of 1.7 and 1.9 points in zero-shot and few-shot scenarios, respectively. The ARG method's effectiveness can be attributed to its ability to train the model on a diverse set of reasoning tasks, enhancing its adaptability and robustness across various contexts. By exposing the model to reasoning tasks of varying lengths and complexities, ARG enables the model to develop a more nuanced understanding and generates more accurate stances.

The experimental findings clearly demonstrate that integrating target knowledge and reasoning generation tasks provides complementary benefits, significantly improving the overall effectiveness of stance detection models. The combination of rich, detailed target knowledge from LLMs and the enhanced reasoning capabilities developed through reasoning generation tasks enables the model to perform at a much higher level. This synergy is particularly valuable in real-world applications where models must handle diverse and complex data efficiently.

**Comparing model performance using external knowledge from Wikipedia *vs*. LLM.** In the third and fourth blocks of Table 2, we compare models incorporating external knowledge from Wikipedia and LLMs. The results show that the approach utilizing LLM target knowledge surpasses the approach employing Wikipedia knowledge across all metrics. Specifically, the model leveraging only LLM target knowledge outperformed the model relying solely on Wikipedia knowledge by 1.2 points in the overall F1 score, 1.4 points in the zero-shot scenario, and 1.2 points in the few-shot scenario. These findings underscore the greater effectiveness of LLM target knowledge in both zero-shot and few-shot scenarios.

Moreover, the inclusion of reasoning generation in the training process further enhances the performance of models utilizing LLM target knowledge. Compared to the model trained on Wikipedia knowledge (third block), the model with LLM target knowledge (fourth block) showed improvements of 1.9 points in the overall F1 score, 1.7 points in the zero-shot scenario, and 2.3 points in the few-shot scenario. This indicates that LLM target knowledge is more effective than Wikipedia knowledge, even when the model is further trained with reasoning generation tasks.

The superior performance of LLM target knowledge can be attributed to its comprehensive and contextually rich information, which enhances the model's understanding of the target. This deep understanding is crucial for accurate stance detection, especially in cases where the target is not explicitly mentioned or is indirectly implied. When combined with reasoning generation tasks, the model's ability to generate coherent and contextually appropriate reasoning based on the input further improves, leading to significant gains in performance.

In conclusion, the experimental results clearly demonstrate that LLM target knowledge, particularly when combined with reasoning generation tasks, provides a substantial performance boost over Wikipedia knowledge. This synergy highlights the importance of

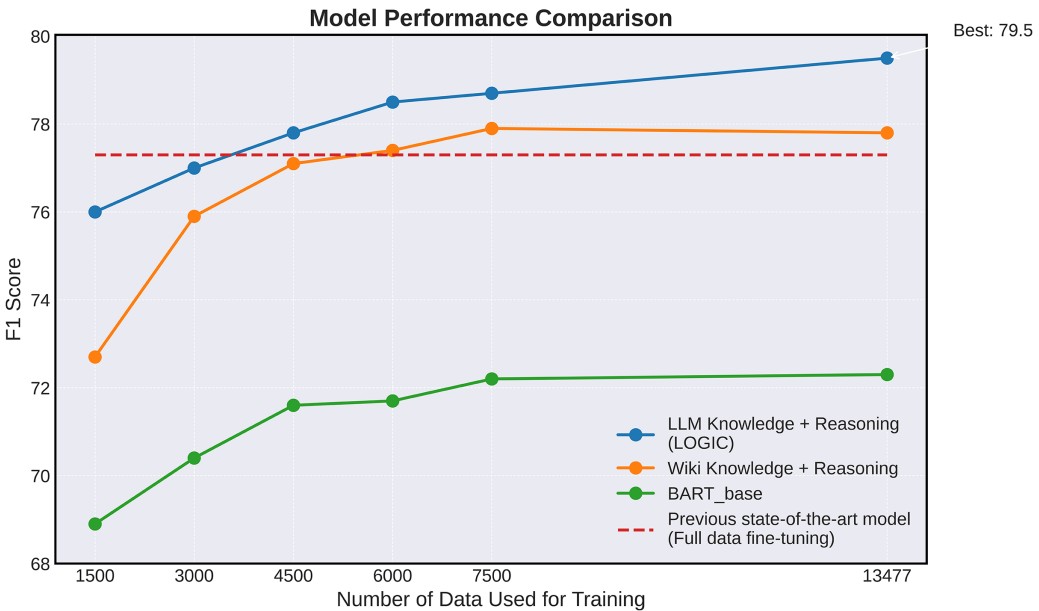

**Figure 4** Graph illustrating the performance of the core models listed in Table 2 trained with varying amounts of training data samples on the VAST test set.

integrating rich, detailed target knowledge from LLMs with advanced reasoning capabilities to achieve state-of-the-art performance in stance detection tasks.

## Evaluating performance based on the amount of data

In this section, we conduct a detailed analysis of how the model's performance varies with changes in the training dataset size. Figure 4 illustrates the impact of the amount of training data on the stance detection performance of the model. Performance was evaluated by averaging the results obtained using five different seeds. Four model configurations were compared and analyzed: a base model exclusively trained on the stance detection task ($BART_{base}$), a model additionally trained on Wikipedia knowledge and reasoning generation (Wikipedia Knowledge + ARG), a model that incorporating knowledge extracted from LLM rather than Wikipedia knowledge (LLM Target Knowledge + ARG), and a model achieving state-of-the-art performance ($BART_{generation}$) (*Wen & Hauptmann, 2023*). Of these, $BART_{generation}$ serves as a baseline, representing the best performance reported by *Wen & Hauptmann (2023)* but it is not evaluated for changes in performance with varying data volumes.

The experimental results demonstrate that our LOGIC model surpasses previous state-of-the-art model in performance, achieving superior results with only 4,500 data samples. This highlights the effectiveness of our approach in delivering high performance even in scenarios with sparse data. This property has important implications for stance detection, particularly in data-limited environments. For a fair comparison with the previous state-of-the-art model, this experiment was conducted using the test set.

|  [0-shot]  |  [24-shot]  |
| --- | --- |
| Predict the expressed stance on a topic from the context of the post and related topics. Please provide your best objective judgment based on the content of the post and your all background knowledge. The output must be expressed as a single number. With just one number. If negative: 0, If positive: 1, If neutral: 2 | Predict the expressed stance on a topic from the context of the post and related topics. Please provide your best objective judgment based on the content of the post and your all background knowledge. The output must be expressed as a single number. With just one number. If negative: 0, If positive: 1, If neutral: 2. I will give you some examples of sets of posts, topics, and correct stance labels to help you make your decision.<br><br>First, we will show examples of existing datasets. The total number of examples is 24. And then at the end you will be provided with the data we need to detect the position. Please refer to the following few shot examples :<br><br>Example 1:<br>"Topic": "street style",<br>"Post": "I truly believe that fashion everywhere is influenced not only by the same people, but by the same websites..."<br>"Stance": 0,<br><br>...<br><br>Example 24:<br><br>...<br><br>And now predict the stance between topic and post from the last following data:<br>Topic: "{topic}"\n  Post: "{post}"\n Stance:` |

**Figure 5  Prompts for stance detection through GPT-3.5 Turbo and GPT-4 Turbo.**

## Performance comparison analysis with existing models

To thoroughly assess our model's stance detection capabilities, we conducted comparisons using the VAST dataset. To ensure a fair comparison against current benchmarks, we used the models specified by *Wen & Hauptmann (2023)*, along with the latest model, Stance Detection *via* Topic-Agnostic and Topic-Aware Embeddings (TATA) (*Hanley & Durumeric, 2023*). This enabled a comprehensive evaluation against a range of state-of-the-art models, including Topic-Grouped Attention (TGA) Net (TGA-Net) (*Allaway & McKeown, 2020*), BERT-GCN (*Lin et al., 2021*), Commonsense Knowledge Enhanced Network (CKE-Net) (*Liu et al., 2021*), Wikipedia Stance Detection BERT (WS-BERT) (*He, Mokhberian & Lerman, 2022*), JointCL (*Liang et al., 2022b*), TATA (*Hanley & Durumeric, 2023*), and BART$_{generation}$ (*Wen & Hauptmann, 2023*), to accurately gauge the performance of our model.

A comparative analysis with existing models on VAST is presented in Table 3. Our methodology achieved the best performance across all scenarios, particularly in the zero-shot scenario, where it outperformed the best existing model by 3.1 points. This demonstrates the superiority of our approach in identifying the stance of an unseen target. Our model matched the best-performing model in terms of few-shot performance, indicating its strong ability to predict stances on previously seen targets. In the overall score, our model outperformed the previous best-performing model by 2.2 points, setting a new state-of-the-art performance on the VAST dataset.

### Performance comparison analysis with GPT-3.5 turbo and GPT-4 turbo

In this study, we compared the performance of our LOGIC model with GPT-3.5 Turbo and GPT-4 Turbo in 0-shot, 12-shot, and 24-shot settings, chosen to evaluate the models' performance across varying levels of example exposure. The 24-shot setting, in particular, was selected to ensure the input length remained within the maximum token limit for GPT-3.5 Turbo. The GPT models were given prompts to predict the stance between a given topic and post, classifying the stance as negative (0), positive (1), or neutral (2). The detailed prompts and performance results are shown in Fig. 5 and Table 4, respectively. Each performance score corresponds to an overall score, as the experimental results are for the entire test set.

Our experiments revealed that GPT-3.5 Turbo achieved its best F1 score of 53.5 in the 24-shot setting, significantly lower than the 79.5 F1 score attained by our LOGIC model. Similarly, GPT-4 Turbo reached its highest F1 score of 65.2 in the 0-shot setting, which also falls short of the LOGIC model's performance.

By examining performance across the 0-shot, 12-shot, and 24-shot settings, we provide a comprehensive evaluation of each model's ability to generalize. While GPT-3.5 Turbo's performance improved slightly as the number of examples increased, it still underperformed compared to the LOGIC model. Interestingly, GPT-4 Turbo's performance decreased as more few-shot examples were introduced, suggesting that providing few-shot examples may hinder its strong zero-shot inference capabilities.

It is important to clarify that the "n-shot" settings here in Table 4 refer to the number of examples provided in the prompt for in-context learning, as is typical for GPT models. In contrast, the zero-shot and few-shot settings mentioned in Tables 2 and 3 refer to whether the stance detection model had encountered the target topic in the training set. This distinction prevents confusion between in-context learning and stance detection training scenarios.

Overall, these results highlight the robustness and superior performance of our LOGIC model in stance detection tasks compared to GPT-3.5 Turbo and GPT-4 Turbo, regardless of the number of in-context examples provided to the GPT models.

## CONCLUSION

In this research, we developed a novel stance detection approach that significantly enhances the reasoning capabilities of SLMs by distilling and utilizing the internal knowledge and reasoning of LLMs as auxiliary learning data. Although this approach inherently relies on the performance of LLMs for knowledge extraction and reasoning distillation, we mitigated the risks of hallucination and bias by structuring the distillation process to focus on explaining the logical relationships between existing labels in the dataset, rather than generating new, uncertain information. The superior performance of the LOGIC model on the VAST dataset validates this approach, with the zero-shot scenario generally outperforming the few-shot scenario. This might seem counterintuitive, but we hypothesize that in the zero-shot scenario, the model can rely more on its internal reasoning capabilities without the interference of potentially noisy or incomplete target information encountered during training. In contrast, the few-shot

scenario may expose the model to partial or insufficient data about the target, leading to suboptimal generalization and reduced accuracy. Furthermore, this superior performance in zero-shot stance detection suggests that the model may offer valuable insights for solving real-world problems where similar zero-shot conditions are often encountered. Beyond stance detection, this methodology holds potential for broader applications involving logical relationship inference between entities. Additionally, the relatively small parameter size of the model makes it well-suited for deployment on resource-constrained devices like smartphones and IoT systems. Future research should focus on optimizing the model for a wider range of NLP tasks while further enhancing efficiency and addressing the potential risks posed by hallucination and bias, particularly in reasoning generation and target knowledge extraction. Hallucinations, where LLMs generate false or misleading information presented as facts, could significantly undermine the accuracy and reliability of stance detection, leading to incorrect conclusions or biased decisions. It is critical to develop strategies to detect and filter out such erroneous information to ensure the model's predictions remain trustworthy, especially in real-world applications.

### Funding
This work was supported by Institute of Information & Communications Technology Planning & Evaluation (IITP) grant funded by the Korea government (MSIT) (2019-0-00004, Development of semi-supervised learning language intelligence technology and Korean tutoring service for foreigners). Also, this work was supported by Institute of Information & Communications Technology Planning & Evaluation (IITP) grant funded by the Korea government (MSIT) (RS-2023-00216011, Development of artificial complex intelligence for conceptually understanding and inferring like human). The funders had no role in study design, data collection and analysis, decision to publish, or preparation of the manuscript.

### Grant Disclosures
The following grant information was disclosed by the authors:
Institute of Information & Communications Technology Planning & Evaluation (IITP).
Korea government (MSIT): 2019-0-00004, RS-2023-00216011.

### Competing Interests
The authors declare that they have no competing interests.

### Author Contributions
- Woojin Lee conceived and designed the experiments, performed the experiments, analyzed the data, performed the computation work, prepared figures and/or tables, authored or reviewed drafts of the article, and approved the final draft.

- Jaewook Lee conceived and designed the experiments, performed the experiments, analyzed the data, performed the computation work, prepared figures and/or tables, authored or reviewed drafts of the article, and approved the final draft.
- Harksoo Kim conceived and designed the experiments, authored or reviewed drafts of the article, and approved the final draft.

## Data Availability

The data is available at Zenodo: Woody Lee. (2024). 10kH/LOGIC: Version 1.1.0 (v1.1.0). Zenodo. https://doi.org/10.5281/zenodo.13150923.

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
