# Peer review of "LOGIC: LLM-originated guidance for internal cognitive improvement of small language models in stance detection"

_PeerJ Computer Science, doi:10.7717/peerj-cs.2585_

## Round 0.1 · original submission · Major Revisions

The paper requires some improvements as indicated by the reviewers. Please, address all the reviewers comments. In particular, the related work section should include recent literature on distillation beyond reasoning, especially for low-resource settings. Expand on data preprocessing steps and provide more detail on the dataset's characteristics (like text types, label sourcing, and splits). Additionally, the presentation of results needs adjustments (definitions, explanations, and visual issues). Finally, address potential biases and hallucinations associated with LLMs and their impact on the methodology's reliability.

·

Basic reporting

The paper is very nicely written and very well structured. The introduction of the paper is outstanding, guiding the reader towards the topic of the paper via a detailed description of the current state of the literature, the shortcomings the proposed approach is trying to overcome and the contributions that the authors propose to achieve state-of-the-art performance in the stance detection task. The language used is very clear throughout the whole paper and – apart from some minor mistakes (e.g., “Overall, These” in line 379 and “less-trust” in line 385) – free of errors. The structure is in line with the PeerJ standards as well as discipline norms. As far as I am aware of, the paper cites the relevant literature on the topic, covering in good detail previous approaches to improve the state-of-the-art performance in stance detection. The presented approaches mainly focus on the VAST dataset that is used also in this paper.

The results are overall well presented, with most terms and concepts well defined and explained. However, I noticed some terms that have not been properly introduced. In most cases, their meaning was obvious from the context, however, I do think that the paper would benefit from properly introducing the following aspects:

1. Across almost all sections of the results, a comparison is made between the performances achieved in a zero-shot and a few-shot setting as well as those resulting overall. While the zero- and few-shot settings are conceptually self-explaining, I neither see where the presented overall setting is defined nor understand what it is supposed to represent. It appears to represent some form of average between the zero- and few-shot settings, but it does not just seem to be the mean between those two settings. Either way, I would highly encourage the authors to explain what the overall setting exactly represents, and to motivate why it is needed, if – as I suspect – it only represents some form of average of the zero- and few-shot settings. Why is there a need to introduce this overall measure instead of just – as the authors do quite well – comparing the zero- and few-shot results?

2. In Figure 2, the authors show sample templates for the different subtasks. While this is very instructive and helpful to understand the inputs going into the models, it would also be interesting to see the text instance corresponding to the example prompts, i.e., the “given post” referenced in the Reasoning Generation part of the same Figure. This would not just be useful to better understand the properties of the dataset (more on that in the section Experimental Design), but also to follow this specific example – e.g., without knowing the post that the templates are filled in for, I cannot (truly understand /) verify whether the “false” stance shown in the Unlikelihood Training example is actually false.

3. In Figure 3, the tokens {false1} and {false2} are used, without having been introduced anywhere else. While I can speculate that they represent the stance labels that are not indicated as the correct stance label for the instance as imposed by the dataset used, I think it would be good to state their exact meaning somewhere.

Experimental design

Building upon the strong introduction and description of the issues seen in the current literature as well as the contributions proposed to overcome then, I think the authors do a very good job in developing and describing their methodology. Both the individual components as well as their interactions and interplay are well motivated and explained.

However, the description of the dataset used for both development of the method and its evaluations and comparison to existing alternatives does not live up to the same standard. While I understand that the (re-)used dataset is a popular, well-known resource in the stance detection literature, I still think some more details in the description of the data – given that it plays such a central role in the whole approach – would be warranted. Concretely, I would like to know what types of texts the dataset contains (e.g., Social media posts? News articles? A mixture? Additionally, I would also be interested in knowing the average length of the text instances in the dataset), how the labels (i.e., the topic/target and the corresponding stance) were produced (Expert annotators? Crowdsourcing?), and how the instances were split into separate training and testing datasets (as well as their corresponding sizes). This all would help to better understand the settings in which the proposed methodology may be expected to perform well and to contextualize the results more precisely.

Connected to this, I think that the sentence on the avoidance of ‘preprocessing of raw data’ reads a bit awkward. From the author comments that have been made available for this review, I understand that this sentence has been added only retrospectively. I could imagine that it works better when placed in the Dataset section and when stressing that the data is not preprocessed, but rather that the different components of the dataset instances are inserted for the different placeholders introduced and explained in the Methodology. The “target knowledge and reasoning by LLMs”, currently described as “treat[ed] as separate datasets”, could then maybe be introduced at the same location proposed above as augmentations of the instances in the existing VAST dataset, which would maybe be a more fitting description of how these dataset were created in the context of the dataset’s instances. These are of course only my personal preferences and suggestions, but I do think that the current description and presentation of the used dataset(s) should be improved.

While the caption of Table 1 explains that the row corresponding to the LOGIC model is highlighted in gray, there is no explanation of which values are printed in bold. I am pretty sure that these are the highest values for each of the performance metrics and settings, but I think stating this explicitly (at least once for Table 1 and in addition pointing out that this convention holds true across all other tables) would benefit the overall presentation of the paper.

When moving through the many results presented in Table 1, the authors reference different “blocks” (e.g., “first block” in line 238) within the table – however, both the rows and columns are visually organized into blocks. Again, in continuation it becomes obvious that what the authors refer to are “blocks of rows”, i.e., different model and component combinations. However, parallel to some of the aspects pointed out above, I think it would improve the immediate understanding of the argumentation if the authors would state the obvious at least once, e.g., by referring to “first block of rows” instead of “first block” when first introducing the unit of blocks in line 238.

While I like the idea of Figure 4 and the reporting of model performance for different sizes of the training dataset in general, I have some major issues with how Figure 4 is currently presented. First of all, I think it is misleading to change the scale of the x-axis by using the same tick-distance for fundamentally different values – while the spaces between the first five ticks represent a change in dataset size of 1,500, the equal space between the last two ticks represents an increase of 5,977 in dataset size. While the authors do not conclude anything from this “warped” x-axis and the effect it might or might not imply – depending on whether collapsing this major increase in training dataset size into the basic tick distance unit or not – it still misrepresents the training dynamics. I would highly encourage the authors to either increase the width of the figure to fully represent the value space without having to collapse the x-axis, or to use any other method to signal this break in the x-axis and to point it out explicitly. Also in the context of Figure 4, I think the section Evaluating Performance Based on the Amount of Data should state explicitly that only the first of the four model configurations actually are evaluated with respect to the increasing amount of training data used, and that the “model achieving state-of-the-art performance” is always using the full amount of training data – i.e., for this model, the evaluation suggested in this section is actually not possible. This is – to a degree – implied in the legend of Figure 4, but I think it should also be pointed out in the main text, especially since the main text currently seems to imply that the amount-of-data comparison is performed on the four models, when – in truth – it is only performed for the first three configurations, which are then compared to the baseline provided by the full-training-data, state-of-the-art model. Being more precise here would actually strengthen the argument made in favor of the proposed methodology, in line with what is laid out in lines 342 and 343.

I cannot find the information about the scenario which was used for LOGIC in Tables 3. I suspect that the reported values for LOGIC represent the Overall scenario that was already used in Table 1, but am not able to verify this. This should be stated explicitly somewhere, especially given that the GPT-3.5/GPT-4 based settings cover both few- and zero-shot setups.

I do not understand why the values for LOGIC in Tables 2 and 3 are not equivalent to those already reported in Table 1, i.e., why are the F1 score values for zero-shot, few-shot and Overall for LOGIC reported in Table 2 not equivalent to the F1 score values for zero-shot, few-shot and Overall for LOGIC in Table 1. If this is because the identical settings for LOGIC have been run multiple times for each of the different tables – in each case for five different random seeds, as reported elsewhere – I would be a bit taken aback by the degree to which the results seem to fluctuate across the different random seeds (1.2 percentage points for zero-shot and 2 [!!!] percentage points for few-shot), and would very much encourage the authors to report the variances in these performance metrics across different runs.
Apart from this handful of issues and concerns I have – mainly with the (re)presentation of results – I think that the experiments are well designed and suitable to support the claims of the authors. Sufficient information is given for the replication of the experiments, and the experiments seem to have been conducted with rigor and on a high technical standard. Sources are cited adequately.

Validity of the findings

I think the whole paper is making a good argument why the proposed contributions are suitable to improve the state-of-the-art in stance detection, and the experimental results support this argument. The conclusions are in good detail and supported by experimental evidence. The whole argumentation is coherent and built-up nicely from Introduction to Conclusion.

In the discussion of the results, however, I am missing a discussion of the (counter-intuitive?) observation that the zero-shot setting performs better than the few-shot setting. At least in my experience, the inclusion of additional “shots” generally leads to improved performances – do the authors have any idea why this does not hold here? Is there any good reasoning why zero-shot should (be expected to) perform better than few-shot?

I am also of the opinion that a single (half) sentence on the limitations (“While these models also have the limitation of relying on the performance of the LLMs themselves to transfer internal knowledge and reasoning power, [..]”) of the proposed approach is by no means sufficient, especially when – as pointed out correctly – relying so heavily on LLM generations that are known for their many shortcomings. In the case of the proposed application, the possible biases introduced through the LLM-generated knowledge and reasoning seems to be of elevated relevance, as is the tendency of LLMs to “make-up” information presented to the user as facts (“hallucinations”). I would very much like to ask the authors to reflect on the question of what these known LLM limitations would mean for their method and application, i.e., whether there would be a problem if the whole stance detection approach is relying on biased or outright false knowledge and flawed reasoning.

Reviewer 2 ·

Basic reporting

The paper is written in clear and professional English throughout, and the structure follows the standards expected for PeerJ. The introduction effectively introduces the subject of stance detection and highlights its significance in social science research, as well as the existing limitations of small language models (SLMs) when using Wikipedia-sourced data. The motivation for improving stance detection through large language models (LLMs) and reasoning distillation is well-articulated.

However, while the background is well-referenced, the related work section could be expanded to include more recent literature on distillation methods beyond reasoning, particularly in low-resource settings. This would further enrich the discussion and place the proposed methodology in a broader context. Additionally, more clarity in distinguishing the contributions from existing approaches would strengthen the novelty of the work.

Experimental design

The experimental design is rigorous and falls within the journal's aims and scope. The methodology is described in detail, including target knowledge extraction, reasoning distillation, and auxiliary learning tasks. The choice of BART as a base model and the use of GPT-3.5 Turbo for reasoning generation are well-justified. The authors also provide sufficient detail regarding the experimental setup, including the computing infrastructure and dataset used (VAST), making replication feasible.

However, the data preprocessing is briefly mentioned but lacks sufficient detail. Given the critical role of data preprocessing in NLP tasks, the authors should provide more information on how the raw data was handled before being input into the models. Additionally, a clearer explanation of the model's hyperparameters and training procedure would be helpful for researchers looking to replicate the results.

Validity of the findings

The findings are valid and well-supported by the experimental results. The authors present a thorough evaluation of their model against existing methods and provide comprehensive metrics (precision, recall, F1) to demonstrate LOGIC's superior performance. The ablation studies and comparison with state-of-the-art models, including GPT-3.5 Turbo and GPT-4 Turbo, are particularly strong aspects of the paper.

The conclusions are well-stated and aligned with the results. The authors also discuss limitations, such as the reliance on the performance of LLMs for target knowledge extraction and reasoning generation. However, the discussion on future directions could be expanded to explore potential applications beyond stance detection and how the model could be further optimized for deployment on smaller devices.

Additional comments

Overall, this paper makes a meaningful contribution to the field of stance detection by addressing the limitations of SLMs and introducing an innovative approach using LLMs for reasoning distillation. The paper is well-structured, with clear arguments and thorough experimental validation. The inclusion of the LLM-generated reasoning data adds a novel layer to the research, and the results on the VAST dataset are promising. I recommend the authors address the issues with data preprocessing and related work to further strengthen the paper.

---

## Round 0.2 · accepted · Accept

Thank you for your contribution to PeerJ Computer Science and for systematically addressing all the reviewers' suggestions. We are satisfied with the revised version of your manuscript and it is now ready to be accepted. Congratulations!

·

Basic reporting

no comment

Experimental design

no comment

Validity of the findings

no comment

Additional comments

I think the authors have done a very good job in revising my minor concerns and have addressed them very sensibly and convincingly. I particularly appreciate the care taken to better introduce the dataset and data processing, to provide additional examples, to "fix" Figure 4, and to clarify some of the conceptual confusion around the different understandings of zero- and few-shot approaches. After having revised the updated version of the manuscript as well as the rebuttal letter, I have no additional concerns or suggestions for further improvement of the work and would be happy to see this published.

Reviewer 2 ·

Basic reporting

The article has substantially improved presentation.

Experimental design

The revision now satisfies my comments regarding experimental design.

Validity of the findings

Given all the efforts made by authors in addressing my points, I believe that the paper can now be accepted.

Additional comments

N/A